# Banana Peel Powder Biosorbent for Removal of Hazardous Organic Pollutants from Wastewater

**DOI:** 10.3390/toxics11080664

**Published:** 2023-08-01

**Authors:** Kelly C. S. Farias, Rita C. A. Guimarães, Karla R. W. Oliveira, Carlos E. D. Nazário, Julio A. P. Ferencz, Heberton Wender

**Affiliations:** 1Nano & Photon Research Group, Laboratory of Nanomaterials and Applied Nanotechnology (LNNA), Institute of Physics, Federal University of Mato Grosso do Sul, Campo Grande 79070-900, MS, Brazil; 2Graduate Program in Health and Development in the Midwest Region, Medical School, Federal University of Mato Grosso do Sul, Campo Grande 79070-900, MS, Brazil; 3Institute of Chemistry, Federal University of Mato Grosso do Sul, Campo Grande 79070-900, MS, Brazil; 4Faculty of Engineering, Architecture, Urbanism, and Geography, Federal University of Mato Grosso do Sul, Campo Grande 79070-900, MS, Brazil

**Keywords:** adsorption, biomass, glyphosate, atrazine, pesticides, dye

## Abstract

Disposing of pollutants in water sources poses risks to human health and the environment, but biosorption has emerged as an eco-friendly, cost-effective, and green alternative for wastewater treatment. This work shows the ability of banana peel powder (BPP) biosorbents for efficient sorption of methylene blue (MB), atrazine, and glyphosate pollutants. The biosorbent highlights several surface chemical functional groups and morphologies containing agglomerated microsized particles and microporous structures. BPP showed a 66% elimination of MB in 60 min, with an adsorption capacity (*q_e_*) of ~33 mg g^−1^, and a combination of film diffusion and chemisorption governed the sorption process. The biosorbent removed 91% and 97% of atrazine and glyphosate pesticides after 120 min, with *q_e_* of 3.26 and 3.02 mg g^−1^, respectively. The glyphosate and atrazine uptake best followed the Elovich and the pseudo-first-order kinetic, respectively, revealing different sorption mechanisms. Our results suggest that BPP is a low-cost biomaterial for green and environmentally friendly wastewater treatment.

## 1. Introduction

Pesticides are essential in modern agriculture, as they increase food production and reduce weeds and pests in plantations [1]. However, overuse does pose severe risks to human health and the environment. The water resources contaminated with these pollutants are a severe problem, originating from the indiscriminate use of these pollutants in agricultural areas and at accidental spill sites [2]. The existence of toxic chemicals, such as pesticides, pharmaceuticals, heavy metals, and dyes, even in low or trace concentrations, has been highlighted by the scientific community because they are very hazardous to the environment, aquatic ecosystems, and human health [3].

Despite the abovementioned concerns, using pesticides on a large scale will likely persist for a long time due to their great importance in agribusiness. Contamination caused by these compounds in aqueous media is recurrent and conventional treatments are ineffective at removing them, generating the need for new treatment technologies to reduce or eliminate their adverse effects [4]. Methods such as adsorption, electrochemical oxidation, advanced oxidation processes, chlorination, bioremediation, ultra-filtration, photocatalysis, and reverse osmosis have been adopted for removing hazardous inorganic/organic pollutants from wastewater [5,6]. Among them, adsorption is a feasible method due to its simplicity, ease of operation, and high removal efficiency [7]. It is also not a destructive method, as adsorption allows the recovery of the adsorbent. Activated carbon, for example, has been extensively studied as an adsorbent, but its high cost limits its large-scale application [6,8]. 

Generally, one can subdivide the adsorption process into chemical adsorption (or chemisorption) and physical adsorption (or physisorption). Chemical adsorption is an irreversible and localized process (which can only occur at active sites) that involves the formation of strong bonds between the molecules and the adsorbate ions on the adsorbent’s surface [9]. In contrast, physical adsorption is a reversible and non-localized process (occurring on the entire surface of the adsorbent), characterized by weak van der Waals forces, π-π and dipole-dipole interactions, and electrostatic forces [7], where the adsorbed substance does not penetrate the solid but remains on its surface.

When the adsorbent is a biological source material, it is called a biosorbent. Biosorption using biomass materials is of considerable interest since it is low-cost, environmentally friendly, biodegradable, and presents interesting physicochemical properties for the adsorption of metals or organic compounds [10]. In some cases, plant biomass has a high surface area and efficiently enables the removal of contaminants by physical-chemical mechanisms similar to activated carbon sorbents [3]. The usefulness of several biosorbent materials comes from a wide variety of organic groups in their surface composition, such as carboxylic acids, phenols, amines, and amides, that can retain metal ions or pollutant molecules through different mechanisms (chelation, ion exchange, and physical adsorption) [11,12]. Materials derived from biomass are also fibrous, allowing more significant contact between the ions in the solution with the active sites of the material. 

Among different and interesting biomass materials, banana peel is an alternative for generating new biosorbents, given its abundance and low cost [13,14,15]. The banana is the world’s fourth most cultivated fruit, and its peel waste is readily available, underutilized, and can be harnessed for commercial applications [16]. Therefore, reusing this material to remove organic pollutants from contaminated water is a process of high commercial interest. The use of dried banana peel powder (BPP) as a biosorbent material has been explored for the decontamination of wastewater, mainly in the removal of textile dyes [17,18,19,20], heavy metals [21,22,23], crude oil [24], phenolic compounds [25], pharmaceuticals [26,27], and pesticides [27,28,29], as has been well documented in two relatively recent reviews [16,30]. Most studies reporting banana peel-based biosorbents have shown efficient biosorbent reuse up to five cycles and above [16,31]. As for all biomass-derived biosorbents, the utilization of banana peel in wastewater treatment applications, while showing promising academic potential, does have some limitations for real-world scenario applications that must be further addressed, including issues related to stability, variability in sorption capacity, scale-up challenges, and regeneration/reusability concerns.

BPP was recently evaluated as a biosorbent for atrazine and ametryne removal from water [27]. The authors observed high adsorption capacities for both pesticides without chemical modifications on the peel surface or pH adjustment. In another study, plantain banana peels (*Musa paradisiaca* L.) were employed as biosorbents for removing the pesticide metribuzin from an aqueous solution, showing a maximum adsorption capacity of 167 mg g^−1^ [28]. Atrazine removal from water using treated banana peels as a biosorbent was also shown to be both pH and temperature dependent. In an optimized condition (60 °C, 100 mL of atrazine @250 ppm, and 1.5 g of BPP pre-treated with H_3_PO_4_), the biosorbent showed a maximum adsorption capacity of 14 mg g^−1^, and both external mass transfer and intraparticle diffusion played essential roles in the adsorption mechanisms [29]. However, works regarding the efficacy of banana peel at different maturation stages as biosorbents for removing pesticides in water are scarce. To our knowledge, adsorption capacity and mechanisms have not been reported in glyphosate pesticides.

Herein, we report *Musa paradisiaca* L. BPP biosorbents prepared at three stages of maturation to remove the hazardous pollutants MB, glyphosate, and atrazine from water, including a deep discussion of the biosorbent adsorption capacity at equilibrium, the kinetic mechanisms, and the uptake efficiency of aqueous organic pollutants. The readily available banana peels were processed to obtain a powdered adsorbent without any chemical treatment, ending in a completely chemical-free, green, and environmentally friendly material. The biosorbent proposed in this study is relatively simple, and it can be applied for real samples in future works regarding the removal of recalcitrant pesticides and other organic pollutants from water bodies.

## 2. Experimental

### 2.1. Materials

Methylene blue (MB, molecular formula: C_16_H_18_N_3_ClS) was purchased from Merck and used as a model water-soluble cationic azo dye. Atrazine (molecular formula: C_8_H_14_ClN_5_), 99% pure, was purchased from Sigma-Aldrich in Brazil, and the commercial glyphosate product (C_3_H_8_NO_5_P) used was the Roundup^®^ Transorb from Monsanto (a soluble concentrate containing 480 g L^−1^ glyphosate, present as 588 g L^−1^ (43.8% *w*/*w*) of the potassium salt of glyphosate). All the reagents were of analytical grade and were used without further purification. All aqueous solutions in this study were prepared using distilled water. 

### 2.2. Banana Peel Powder Preparation

The banana peel samples were obtained from the Mato Grosso do Sul Supply Center (CEASA) in Campo Grande, MS/Brazil. The fruits were cleaned with running water, immersed in a hypochlorite solution (10 mL L^−1^) for 10 min, and manually peeled with a stainless-steel knife. The banana peels at three stages of ripening (green, semi-ripe, and ripe) were dehydrated in a forced air oven at 40 °C until the mass stabilized, crushed in a mill to obtain the powders (BPP), sieved through a 40-mesh granulometric sieve, and, finally, vacuum packed in polypropylene plastic bags. 

### 2.3. Characterization of the Biosorbent

The BPP’s scanning electron microscopy (SEM) images were acquired by dispersing the powder on a double-sided carbon tape fixed to the stub and then covered by a thin layer of gold by sputtering (300 s under 0.2 Torr pressure in a Desk III model Denton Vacuum evaporator). The images were obtained using an SEM microscope (JEOL, model JSM-6380 LV) under the three-pronged conditions: an accelerating voltage of 15 kV, a work distance of ~8 mm, and a secondary electron detector. Fourier-transformed infrared spectroscopy (FTIR) measurements were performed in the attenuated total reflectance (ATR) mode, with 20 sweeps for each sample in the range of 4000 to 500 cm^−1^ using a Spectrum 100 (PerkinElmer, Waltham, MA, USA) equipment. Thermogravimetric analysis (TG-DTG) and differential scanning calorimetry (DSC) were performed under an N_2_ atmosphere using a simultaneous thermal analyzer Netzsch STA 449 F3 Jupiter^®^ in the temperature range of 25–600 °C and with a scan rate of 5 °C/min.

### 2.4. Adsorption Experiments

#### 2.4.1. Biosorption of the Methylene Blue Azo Dye

Initially, we selected MB as a pollutant organic compound model for the preliminary study of the adsorption capacity of biosorbents. In the tests, 50 mg of BPP was added to 50 mL of a solution containing MB dye in an initial concentration of 50 mg L^−1^. The kinetic tests were performed for 60 min, under vigorous agitation, with the collection of 500 µL aliquots at predetermined time intervals. After collection, the samples were centrifuged (~8000 rpm) for 10 min to remove the biosorbent, and the supernatant was analyzed by UV-Vis absorption spectrophotometry by monitoring MB maximum peak at 664 nm. All experiments were carried out in triplicate, and the percentage of removal of the dye, the amount of dye adsorbed at time *t*, $q_{t}$ (mg g^−1^), and the adsorption capacity at equilibrium $q_{e}$ (mg g^−1^) were calculated by the following equations, respectively:(1)$$\%Removal=\left(\frac{C_{0}-C_{t}}{C_{0}}\right)\times 100\;$$
(2)$$q_{t}=\left(\frac{C_{0}-C_{t}}{m}\right)\times V$$
(3)$$q_{e}=\left(\frac{C_{0}-C_{e}}{m}\right)\times V\;$$
where $C_{0}$, $C_{e}$ and $C_{t}$ (mg L^−1^) are the initial dye concentration, the dye concentration at equilibrium, and at time *t*, respectively, where *V* (L) is the volume of the dye solution and m (g) is the mass of the adsorbent. 

#### 2.4.2. Biosorption of the Glyphosate and Atrazine Pesticides

The adsorption properties of the biosorbents were also evaluated against the glyphosate and atrazine pesticides in batch experiments, where an aqueous solution of atrazine (or glyphosate) at a concentration of 20 mg L^−1^ was separately prepared (pH~5.5). In each experiment, a certain amount of the biosorbent (10, 30, 50, and 60 mg) was added to 10 mL of the pesticide solution in 50 mL beakers. The mixture was vigorously stirred for 12 h to achieve equilibrium under dark conditions. Kinetic studies were performed for 120 min for both pesticides using 60 mg of the BPP in the semi-ripe maturation stage, with the collection of 500 µL aliquots at predetermined time intervals. For pesticide quantification, the solutions were immediately filtered through a 0.45 µm membrane filter connected to a syringe and analyzed using a Shimadzu HPLC equipment model prominence 20A, equipped with a quaternary pump, automatic sampler, and a DAD detector. The chromatographic method was developed and optimized for each pesticide. For atrazine, the conditions obtained as ideal were methanol/water pH 3.0 (90:10 *v*/*v*) as the mobile phase; Eclipse C18 150 cm × 4.6 mm as the chromatographic column; 4.5 µm and a wavelength of 240 nm; a flow of 0.6 mL min^−1^; an oven temperature of 35 °C; an analysis time of 6 min; and an injection volume of 20 μL. For glyphosate, the parameters were the same, except 70:30 *v*/*v* as the mobile phase, a wavelength of 230 nm, a flow of 0.8 mL min^−1^, and an analysis time of 3 min. All experiments were carried out in triplicate, and % of removal, $q_{t}$, and $q_{e}$ were calculated using Equations (1)–(3).

#### 2.4.3. Kinetic Models of Adsorption

The dye and pesticide adsorption processes were evaluated using the Avrami fractional-order, pseudo-first-order, pseudo-second-order, pseudo-general-order, Elovich, and intraparticle diffusion models to investigate the adsorption mechanism and the main adsorption process steps. The kinetic models were fitted using nonlinear fitting. Table 1 presents the equation related to each model, where $k_{f}$ (min^−1^), $k_{s}$ (g mg^−1^ min^−1^), and $k_{G}^{}$ (min^−1^) are the rate constants of the pseudo-first, pseudo-second, and pseudo-general order models, respectively; t is time in min;$\;q_{t}$ is the amount of adsorbate adsorbed at time t (mg g^−1^); and $h_{0}$ (mg g^−1^ min^−1^) is the initial sorption rate. For the pseudo-general model, n represents the kinetic order of adsorption. In the case of the fractional-order (Avrami) model, $k_{AV}$ (min^−1^) and $n_{AV}$ are the rate constant and the fractional adsorption order, respectively. In the Elovich chemisorption model, α is the initial sorption rate (mg g^−1^ min^−1^), and β (g mg^−1^) is the desorption constant related to the magnitude of surface coverage and activation energy. $k_{id\left(1,2,3\right)}^{}$ (mg g^−1^ min^1/2^) represents the rate constant of intraparticle transport, and C (mg g^−1^) is the boundary layer (film diffusion). Detailed information regarding these kinetic models can be found in the literature [32,33,34,35,36].

## 3. Results and Discussion

### 3.1. Characterization of the Biosorbents

Figure 1A shows the optical images obtained from the fruits and the obtained BPP at the three ripening stages studied in this work. According to the literature, the maturation stages were classified as green, semi-ripe, and ripe [37]. After drying and crushing the banana peels, the powders showed different colors, changing from brown to dark brown as the ripening degree increased (Figure 1A). The calculated yield based on the kilogram weight of the peel was 11.3, 12.8, and 17.9% for green, semi-ripe, and ripe BPP, respectively. Figure 1B shows the FTIR spectra of the BPP for the three ripening stages. The band at around 3277 cm^−1^ can be attributed to the axial stretching vibration of the O–H of the hydroxyl groups characteristic of lignin, cellulose, hemicellulose, and adsorbed water, and the band at 2919 and 2850 cm^−1^ can be attributed to the symmetrical and asymmetrical C–H stretching of methoxy groups in lignin, cellulose, and hemicellulose [20,38]. The band around 1736 cm^−1^ is a signature of C=O stretching in carbonyl or carboxyl groups [20,39]. The band around 1588 cm^−1^ is due to C–O or C=C vibrations. The bands in the 1320–1375 cm^−1^ spectra interval correspond to the phenols’ angular C–O–H deformations and the C–O stretching vibrations in carboxylate (alkanes and alkyl groups) [20]. For the spectral interval 950–1100 cm^−1^, the observed absorption band may indicate the presence of the O–H deformation vibration, the C–O stretching vibration of alcoholic groups, aliphatic ethers, and a β-glycosidic bond present both in cellulose and hemicellulose [20]. Therefore, the FTIR spectra analysis suggests that the as-prepared BPP has elevated potential to conduct the biosorption of cationic organic molecules due to many functional groups in the sample composition [16,40].

We investigated the morphological properties of the BPP samples for the three chosen ripening stages through SEM images (Figure 2A–C). The images show an irregular morphology evidencing the presence of agglomerated and microsized particles with smooth surfaces as well as microporous structures. The BPP showed more agglomerated particles in the ripe stage than in the green and semi-ripe maturation stages, which presented similar morphological features. 

Figure 3 shows BPP samples’ TG, DTG, and DSC curves obtained at different ripening stages. Typical characteristics of thermal degradation of biomass-based materials are observed, which correspond to the release of adsorbed water from moisture up to 100 °C, the release of structural water, and the decomposition of several constituents in the range of 150 to 450 °C, such as hemicellulose, cellulose, lignin, pectin, glucose, starch and protolignin [41,42]. For this temperature interval, all volatile and non-volatile gases (H_2_, CO_2_, CO, CH_4_) are expected to be eliminated from the biomass [43]. The total mass loss at 600 °C was ~70% for all samples. The DTG made it possible to identify the temperature at which the mass losses occurred accurately. Notably, TG/DTG curves at all maturation stages showed the release of adsorbed water at 60 °C and remarkable features of mass loss in the 150–450 °C interval. For the green BPP (Figure 3A), three peaks were observed between 150 and 450 °C, i.e., the main peak centered near 275 °C and two small shoulders near 220 and 355 °C. This last minor feature at 355 °C is present in all the ripening stages, and it slowly intensifies as the ripening process proceeds. However, the peak at 220 °C divides into two peaks for semi-ripe BPP (Figure 3B), which are centered at 150 and 180 °C and become more pronounced and sharper for the ripe BPP (Figure 3C). Therefore, the samples showed distinct mass loss processes for the different ripening stages studied, with more defined and less-overlapped steps for the ripe BPP. It is important to note that TG/DTG curves have been shown to be an analytical tool able to separate and distinguish banana ripeness. All of these observed mass losses also follow the events in the DSC curves, where it is possible to highlight an endothermic peak between 23 and 75 °C, corresponding to the release of water and the characteristic exothermic signs of the thermal decomposition of the biosorbents. 

### 3.2. Adsorption Efficiency Evaluation

#### 3.2.1. BPP as Biosorbent for Methylene Blue Removal

The BPP biosorbent efficiency for MB removal was evaluated at a fixed pH of 5.5 and an ambient temperature (23 °C). We first assessed the effect of the contact time of the pollutant with the biosorbent and then the dynamic process of sorption. Figure 4A,B show MB’s removal percentage and adsorption capacity. The BPP biosorbents showed similar adsorption properties for all stages of maturation studied; however, although we cannot find a statistical difference, a minor trend is present with a higher adsorption ability for the semi-ripe BPP, reaching approximately 66% of MB removal after 60 min. The MB adsorption process proved fast, and about half of the initial concentration disappeared from the solution in the first 10 min of contact with the BPP biosorbents. The semi-ripe BPP biosorbent showed the highest adsorption capacity by removing approximately 33 mg g^−1^ of the dye. It is essential to highlight that even though it comes close, the equilibrium seems to be not ultimately achieved after 60 min. 

We used different kinetic models to the obtained data to explore more valuable information regarding the mechanism of sorption and the potential rate-controlling steps, such as mass transport and chemical reaction processes. The calculated parameters from each model and the corresponding fit curves can be seen in Table 2 and Appendix A. Considering the entire contact time, the best-fitting one was the pseudo-second-order model since it presented the highest correlation coefficient $R_{adj}^{2}$ compared to the pseudo-first-order one. The pseudo-general-order model resulted in n~2 and similar $R_{adj}^{2}$ (Table 2), which corroborates the previous finding. The calculated maximum uptake capacities were 28.94, 31.17, and 30.42 mg g^−1^ for green, semi-ripe, and ripe BPP, respectively. These values are very close to the respective experimental $q_{e}$ of 30.08, 33.09, and 31.86 mg g^−1^, and, therefore, it is possible to infer that the primary sorption mechanism of MB sorbate is chemisorption. Hence, the fast MB removal is due to the binding between the dye and the functional groups on the biosorbent surface [6]. The initial adsorption rate $h_{0}$ from the pseudo-second-order model at ambient temperature was near 24.09 mg g^−1^ min^−1^ for green and semi-ripe BPP biosorbents, and it diminished to almost 18.27 mg g^−1^ min^−1^ when the ripening process was completed. 

Generally, chemisorption is described by the pseudo-second-order and the Elovich models. The former model embodies covalent forces and ion exchange by the sharing/exchange of electrons between the adsorbate and the adsorbent, and the latter is indicated for sorbent materials that contain a heterogeneous surface [44]. Both models are indistinguishable under certain conditions and are frequently used to describe the kinetic of chemisorption [45]. The Elovich model was applied to our data, and it could better describe the obtained MB chemisorption kinetics compared to the pseudo-second-order one. The initial sorption rate α was higher for BPP in the green stage and decreased as the ripening process proceeded but with a similar desorption constant β. It is important to note that desorption constant is exceedingly small compared to the initial sorption rate.

It is well established that the sorption process may involve several steps, including (i) transport of the solute molecules from the bulk aqueous phase to the surface of the solid particles (film diffusion); (ii) diffusion of the solute into the interior of the sorbent pores where the adsorption becomes intraparticle controlled, which is usually a slow process [32]; and (iii) surface chemical reaction or complex formation, where the interaction of molecules of adsorbate with the surface functional groups of adsorbent takes place [44]. The intraparticle diffusion model was used to evaluate these stages. The results could be divided into three parts, with a satisfactory correlation factor, indicating that three steps are present in the kinetics of the biosorption process (Appendix A). In the first step, which relates to the external mass transfer (film diffusion) of dye to the biosorbent surface [46], the semi-ripe BPP presented a higher $k_{id1}$ and, therefore, a faster film diffusion process. However, the $k_{id2}$ was higher for the green BPP, showing that the mass transfer inside the biosorbent through intraparticle or pore diffusion is significantly more efficient for the green BPP compared to the other maturation stages. Finally, the third step, when adsorption approaches the equilibrium, which corresponds to diffusion processes related to chemical reactions or complex formation, shows no considerable differences, and this means that the kinetic is similar for the BPP in the three studied maturations stages.

For all the BPP maturation stages, the diffusion rate was high in the initial steps and decreased as time proceeded, which shows that the adsorption rate is mainly governed by film diffusion at the early stage of the removal of MB dye. Comparing the y-intercept values (*C*_1_–*C*_3_) of BPP biosorbent for the three maturation stages, one can see that higher values were obtained for semi-ripe BPP, validating the more significant boundary layer effect and an enhanced film diffusion for BPP at this maturation stage [33,47]. Interestingly, at all of the studied maturation stages, the linear plots do not pass through the origin, which means intraparticle diffusion is not the only rate-limiting step [33]. Therefore, one hypothesis is that sorption combines film diffusion and chemisorption. The diffusion-chemisorption model was also evaluated to validate this hypothesis [44]. This model considers that chemisorption and film diffusion control the adsorption process, where *K_DC_* is the diffusion-chemisorption rate constant, as expressed in Equation (11):(11)t0.5qt=1KDC+1qet0.5

Appendix A present the fitted curves and the calculated parameters for MB adsorption at the BPP biosorbents for the three maturation stages. A more than 99% correlation coefficient was obtained for all of the BPP maturation stages, with *K_DC_* values of 32.76, 31.40, and 26.25 for the green, semi-ripe, and ripe BPP, respectively. Moreover, the calculated $q_{e}$ values are remarkably close to the experimental results. Therefore, by confronting the best-fitted kinetic models evaluated in this work, it is possible to infer that the overall mechanism of adsorption of the MB dye onto BPP is a multi-step process that includes both physisorption and chemisorption.

#### 3.2.2. Biosorption of Atrazine and Glyphosate 

Figure 5A,B show the total removal percentage of atrazine and glyphosate adsorbed at equilibrium for different BPP biosorbent dosages. It is possible to verify that the adsorption efficiency follows the order semi-ripe > ripe > green for both pesticides. The adsorption efficiency also improved with the increase in the biosorbent mass, which can be attributed to the increased number of active sites. In addition, the total removal is higher for glyphosate than atrazine, where 98% and 93% removal efficiencies were achieved using 60 mg of biosorbent, respectively.

Based on the results obtained at the equilibrium, the adsorption (removal percentage and $q_{t}$) as a function of time was only carried out for semi-ripe BPP and using 60 mg of biosorbent, as shown in Figure 5C,D. The concentration of the pesticides decreased for longer contact times, suggesting the occurrence of adsorption in the solid phase. Atrazine was adsorbed more slowly in the first 15 min than glyphosate, followed by a sudden increase in the adsorption rate, reaching a faster equilibrium at nearly 60 min of reaction (Figure 5C). On the other hand, the uptake of glyphosate was relatively quick in the initial phase of the adsorption, which can be attributed to the accelerated instantaneous external diffusion, as has been described before [29], and which gradually decreased over time as it approached equilibrium at near 90 min (Figure 5D). The amount of dye adsorbed ($q_{t})$ increased over time following the same trend, as observed for the total removal percentage, with $q_{e}$ values of 3.02 and 3.26 mg g^−1^ for atrazine and glyphosate, respectively. 

Different models were investigated and adjusted to the experimental data to understand kinetic behavior better, as presented in Table 3. For atrazine, the Avrami fractional-order showed the highest correlation coefficient; however, the kinetic order is not a fractional number ($n_{AV}$ is close to the unity), and the model converges to the pseudo-first-order model (see Table 1). We then assume the biosorption kinetics of atrazine to follow the pseudo-first-order, where calculated $q_{e}$ = 3.46 mg g^−1^ best approximated the experimental value of 3.02 mg g^−1^ with a satisfactory correlation factor of 93.66%. For glyphosate, the pseudo-second-order model showed the best fit compared to the pseudo-first-order one, with a correlation factor of 97.76%, a calculated $q_{e}$ = 3.99 mg g^−1^ close to the experimental value (3.26 mg g^−1^), and $h_{0}\;$= 0.15 mg g^−1^ min^−1^. The general-order model ends up with a remarkably similar result to the pseudo-second-order kinetic for the glyphosate case; however, as observed for the MB dye, the Elovich equation was fitted with a higher correlation coefficient than the pseudo-second-order model. Therefore, we conclude that the adsorption mechanism of glyphosate is better described by chemisorption through the Elovich model. Interestingly, the model’s initial sorption rate and desorption constant are higher than that obtained for atrazine, showing that glyphosate is initially favored to chemisorb at the BPP particle surfaces. It is noteworthy that the diffusion-chemisorption model did not fit perfectly in the data as it did in the case of the MB dye, with a correlation factor $R_{adj}^{2}$ < 0.7 for both pesticides.

The intraparticle diffusion model was also investigated. Unlike the results that were observed for the MB, the model could only be satisfactorily divided into two parts, indicating that only two well-defined steps are present in the adsorption processes of the pesticides (Appendix A and Table 3). In this case, the diffusion process showed a higher $k_{id1}$ value for atrazine compared to glyphosate, suggesting that the instantaneous surface sorption step is faster in the former case [46]. Moreover, for each of the pesticides, the $k_{id1}$ is far higher than the $k_{id2}$, indicating that the film diffusion predominates in the sorption mechanism, and that the mass transfer inside the biosorbent through intraparticle or pore diffusion is present but less significant. Although rarely described [36], the y-intercept values (*C*_1_ and *C*_2_, in our case) of the intraparticle equation are related to the boundary layer effect, and they showed higher values for the glyphosate pesticide, with an initial negative value for atrazine. A negative value can be explained by the combined effects of film diffusion and surface reaction control [36,48]. As for MB, the linear plots (Equation (10) obtained for the pesticides do not pass through the origin, meaning the intraparticle diffusion is not the only rate-limiting step [33].

Therefore, the kinetic process for the BPP biosorbent has shown itself to be highly dependent on the physicochemical properties of the contaminant, and the interactions with the biosorbent differed between the pesticides studied. It is essential to highlight that the biosorbent could uptake more dye molecules in comparison to the pesticides since the calculated $q_{e}$ were about nine times less for the latter. However, this is the first report regarding glyphosate removal using BPP as a biosorbent with an impressive uptake of 3.26 mg g^−1^ under the studied conditions. For atrazine, two previous studies have applied banana peels as a biosorbent [27,29]. However, we show that our untreated BPP are competitive biomaterials and so far deserve attention, especially regarding the low amount of biosorbent employed in this work and the fact that no pretreatment in acid solution was applied. Since it is a cheap and relatively abundant material that typically decomposes in nature, we have shown herein that BPP is a candidate for wastewater treatment for the removal of hazardous dyes, pesticides, and other organic or inorganic pollutants due to its abundant availability of chemical functional groups that provide a bridge to the sorption of target pollutant molecules.

## 4. Conclusions

In three stages of maturation, banana (*Musa paradisiaca* L.) peel powder was evaluated as a biosorbent for removing MB dye and glyphosate and atrazine pesticides from water. The BPP was thoroughly characterized, revealing the presence of many functional groups on the surface and an irregular morphology encompassing agglomerated and microsized particles with smooth surfaces (starch) and microporous structures. The BPP showed distinct mass loss processes with more pronounced and less overlapped steps for the ripe BPP, and TG/DTG curves have been shown to be a precise analytical tool able to separate and distinguish bananas at different ripening stages. The biosorption properties of BPP were initially evaluated against the MB dye and showed a high adsorption capacity of ~33 mg g^−1^ for the semi-ripe stage, with a fast evolution in time where ~50% of dye was absorbed in 10 min, achieving 66% in 60 min. The primary sorption mechanism of MB sorbate is chemisorption, and the fast MB removal is due to the binding between the dye and the functional groups on the biosorbent surface. However, physisorption is not discarded, as confirmed using the diffusion-chemisorption model. Therefore, the overall mechanism of adsorption of the MB dye onto BPP is a multi-step process that includes both physisorption and chemisorption.

Moreover, the BPP biosorbents were evaluated against atrazine and glyphosate, showing adsorption efficiencies in the following order for both pesticides: semi-ripe > ripe > green. The pesticide sorption efficiency improved with the increase in the amount of biosorbent, showing higher total uptake of glyphosate compared to atrazine, with 98% and 93% removal and a maximum adsorption capacity of 3.26 and 3.02 mg g^−1^, respectively. The biosorption of glyphosate and atrazine followed the Elovich and the pseudo-first-order kinetics, respectively. Therefore, without any chemical treatment, the low-cost biosorbent proposed herein is an entirely chemical-free, green, and environmentally friendly biomaterial for removing recalcitrant pesticides and other organic pollutants from water bodies.

## Figures and Tables

**Figure 1 toxics-11-00664-f001:**
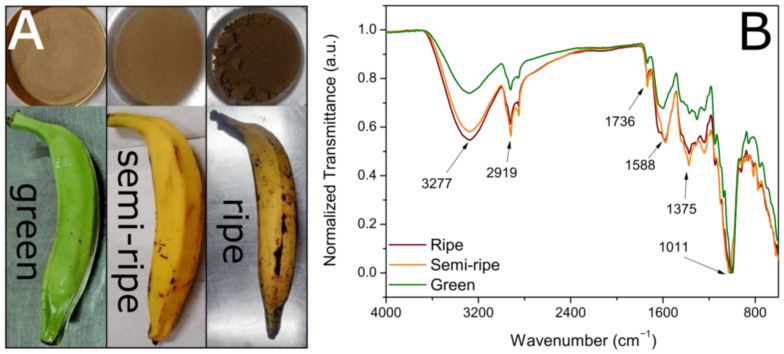
Optical images of the banana fruits and the obtained BPP at the different ripeness stages (**A**) and FTIR spectra of the BPP samples (**B**).

**Figure 2 toxics-11-00664-f002:**
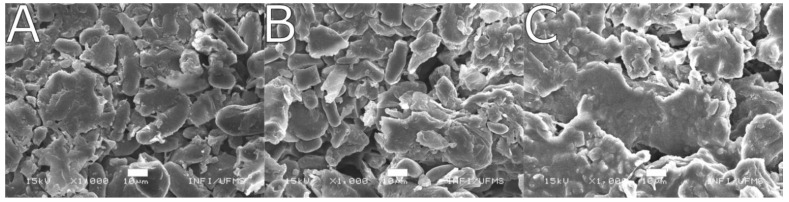
SEM images of green (**A**), semi-ripe (**B**), and ripe (**C**) BPP. Scale bars correspond to 10 µm.

**Figure 3 toxics-11-00664-f003:**
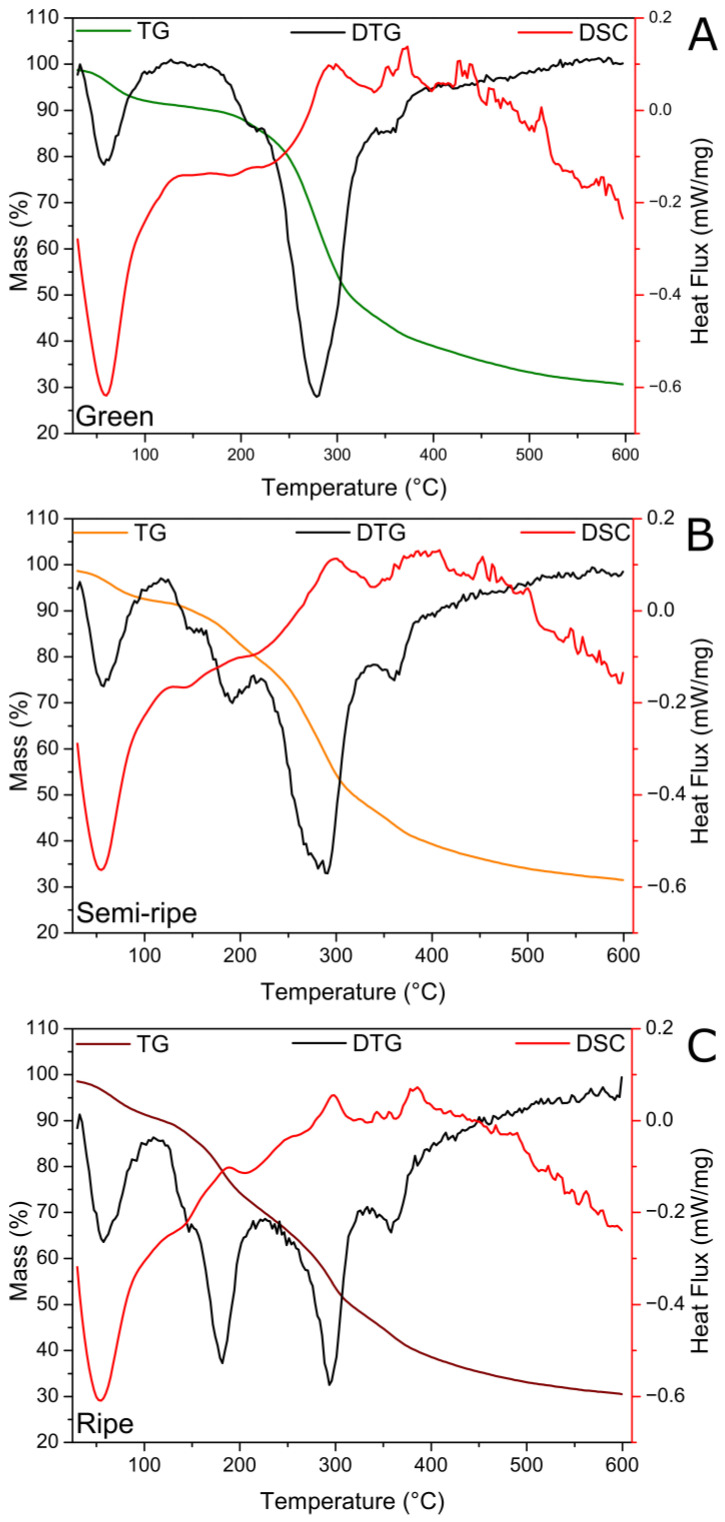
TG, DTG, and DSC curves of the green (**A**), semi-ripe (**B**), and ripe (**C**) BPP biosorbents. The measurements were performed under an N_2_ atmosphere with a 5 °C/min heating rate. The DTG scale was normalized and omitted for easy visualization of the superposed graphs.

**Figure 4 toxics-11-00664-f004:**
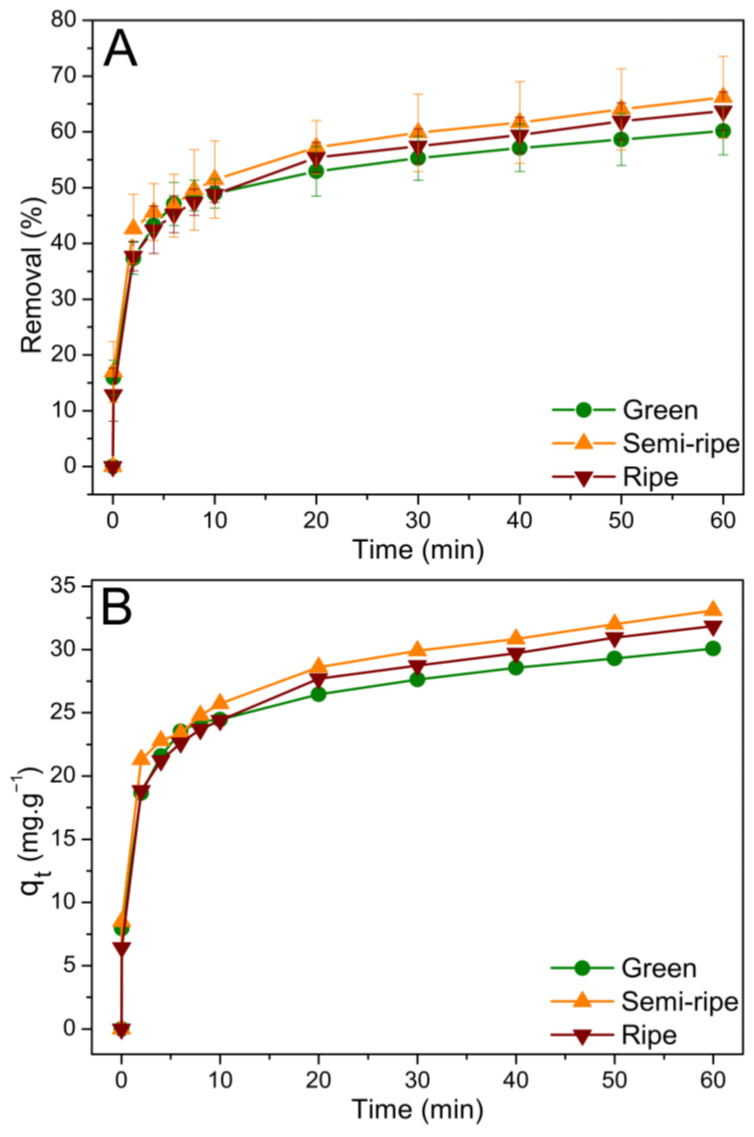
Removal (%) of methylene blue dye (**A**) and adsorption capacity of the BPP biosorbents obtained at different ripening stages (**B**). Conditions: 50 mg of BPP; 50 mL MB dye solution at C_0_ = 50 mg L^−1^.

**Figure 5 toxics-11-00664-f005:**
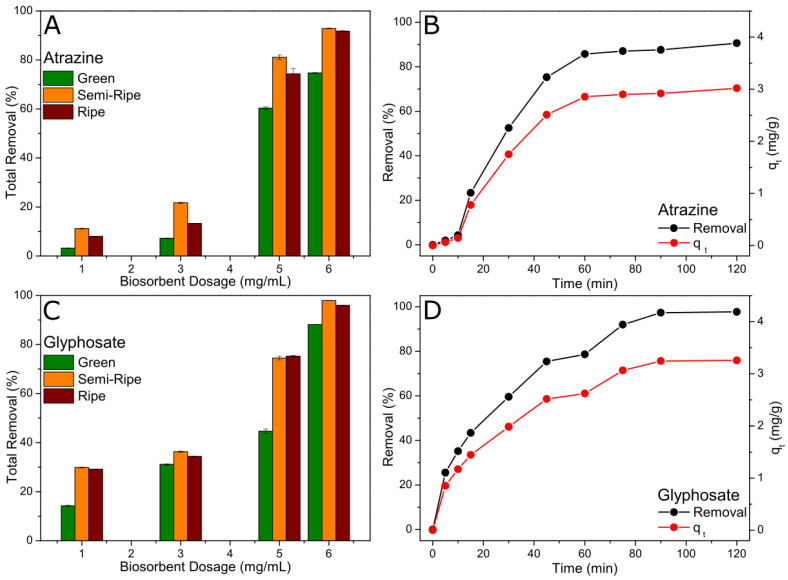
Total removal of atrazine (**A**) and glyphosate (**B**) for different biosorbent dosages after 12 h at the following conditions: 10 mL of pesticide at 20 mg L^−1^ and in contact with 10, 30, 50, and 60 mg of BPP. Percentual removal (left, black) and amount adsorbed at time *t* (right, red) of atrazine (**C**) and glyphosate (**D**) pesticides at the following conditions: 60 mg of semi-ripe BPP and 10 mL of pesticide at 20 mg L^−1^.

**Table 1 toxics-11-00664-t001:** Kinetic adsorption models and their respective nonlinear equations.

Kinetic Model	Nonlinear Equation	Equation Number
Pseudo-first-order	$q_{t}=q_{e}\left\{1-\exp\left(-k_{f}t\right)\right\}$	4
Pseudo-second-order	$q_{t}=\frac{k_{s}q_{e}^{2}t}{1+q_{e}k_{s}t}$	5
	$h_{0}=k_{s}q_{e}^{2}$	6
Pseudo-general-order	$q_{t}=q_{e}-\frac{q_{e}}{{\left(k_{G}^{}{\left(q_{e}^{}\right)}^{n-1}t\left(n-1\right)+1\right)}^{1/\left(1-n\right)}}$	7
Avrami fractional-order	$q_{t}=q_{e}\{1-\exp{\left[-\left(k_{AV}t\right)\right]}^{n_{AV}}\}$	8
Elovich	$q_{t}=\frac{1}{\beta }ln\left(\alpha \beta \right)+\frac{1}{\beta }ln\left(t\right)$	9
Intraparticle diffusion	$q_{t}=k_{id\left(1,2,3\right)}^{}\sqrt{t}+C$	10

**Table 2 toxics-11-00664-t002:** Kinetic parameters for MB removal using green, semi-ripe, and ripe banana peel powders as biosorbents.

Model Parameter	Ripeness Stage		
	Green	Semi-Ripe	Ripe
**Fractional-order**			
$k_{AV}\left(\min^{-1}\right)$	0.24 ± 0.01	0.25 ± 0.01	0.28 ± 1.87 × 10^−6^
$q_{e}\left({\mathrm{mg}\;g}^{-1}\right)$	27.24 ± 1.67 × 10^−8^	29.15 ± 1.56	28.53 ± 1.37
$n_{AV}$	1.85	1.83	1.63
$R_{adj}^{2}$	0.8793	0.8793	0.8839
Residual sum of squares	10.09	10.09	11.19
**Pseudo-first-order**			
$k_{f}\left(\min^{-1}\right)$	0.46 ± 0.11	0.45 ± 1.13	0.35 ± 0.08
$q_{e}\left({\mathrm{mg}\;g}^{-1}\right)$	27.24 ± 1.18	29.15 ± 1.48	28.53 ± 1.30
$R_{adj}^{2}$	0.8914	0.8526	0.8955
Residual sum of squares	9.08	14.41	10.07
**Pseudo-second-order**			
$k_{S}\;\left({g\;\mathrm{mg}}^{-1}{\;\min}^{-1}\right)$	0.03 ± 0.01	0.03 ± 0.01	0.02 ± 0.01
$q_{e}\left({\mathrm{mg}\;g}^{-1}\right)$	28.92 ± 0.19	31.17 ± 1.48	30.62 ± 1.21
$h_{0}\left({\mathrm{mg}\;g}^{-1}\min^{-1}\right)$	24.11 ± 0.04	24.09 ± 0.04	18.27 ± 0.02
$R_{adj}^{2}$	0.9343	0.9152	0.9482
Residual sum of squares	5.49	8.29	5.00
**Pseudo general-order**			
$k_{G}\;\left(\min^{-1}\right)$	0.027 ± 0.001	0.025 ± 0.001	0.019 ± 0.001
$q_{e}\left({\mathrm{mg}\;g}^{-1}\right)$	28.95 ± 1.19	31.17 ± 0.01	30.43 ± 1.01
$n_{}$	2.02 ± 0.01	2.03 ± 0.10	2.03 ± 0.10
$R_{adj}^{2}$	0.9347	0.9233	0.9485
Residual sum of squares	5.46	7.50	5.00
**Elovich**			
$\alpha \;\left({\mathrm{mg}\;g}^{-1}\right)$	798.53 ± 108.62	735.74 ± 127.11	373.85 ± 39.95
$\beta \;\left({g\;\mathrm{mg}}^{-1}\right)$	0.32 ± 0.01	0.29 ± 0.01	0.28 ± 0.01
$R_{adj}^{2}$	0.9982	0.9969	0.9983
Residual sum of squares	0.15	0.31	0.16
**Intraparticle diffusion**			
$k_{id1}\left({\mathrm{mg}\;g}^{-1}\min^{-0.5}\right)$	11.80 ± 3.54	13.64 ± 3.62	12.37 ± 2.44
*C*_1_ (${\mathrm{mg}\;g}^{-1}$)	2.43 ± 2.92	2.49 ± 2.99	1.68 ± 2.02
$R_{adj}^{2}$	0.8349	0.8687	0.9249
Residual sum of squares	14.47	15.10	6.89
$k_{id2}\left({\mathrm{mg}\;g}^{-1}\min^{-0.5}\right)$	4.09 ± 0.45	2.42 ± 0.10	2.82 ± 0.17
*C*_2_ (${\mathrm{mg}\;g}^{-1}$)	13.14 ± 1.00	17.86 ± 0.30	15.40 ± 0.48
$R_{adj}^{2}$	0.9647	0.9910	0.9828
Residual sum of squares	0.45	0.24	0.62
$k_{id3}\left({\mathrm{mg}\;g}^{-1}\min^{-0.5}\right)$	1.20 ± 0.04	1.35 ± 0.06	1.29 ± 0.07
*C*_3_ (${\mathrm{mg}\;g}^{-1}$)	20.91 ± 0.20	22.48 ± 0.35	21.80 ± 0.45
$R_{adj}^{2}$	0.9948	0.9935	0.9879
Residual sum of squares	0.14	0.06	0.10

**Table 3 toxics-11-00664-t003:** Kinect parameters for glyphosate and atrazine removal using semi-ripe banana peel powder biosorbents.

Model Parameter	Pesticide	
	Glyphosate	Atrazine
**Fractional-order**		
$k_{AV}\left(\min^{-1}\right)$	0.02608 ± 0.01	0.02270 ± 0.01
$q_{e}\left({\mathrm{mg}\;g}^{-1}\right)$	3.22 ± 0.01	3.46 ± 0.15
$n_{AV}$	1.41 ± 0.10	0.99 ± 0.01
$R_{adj}^{2}$	0.9609	0.9446
Residual sum of squares	0.03	0.09
**Pseudo-first-order**		
$k_{f}\left(\min^{-1}\right)$	0.04 ± 0.01	0.02 ± 0.01
$q_{e}\left({\mathrm{mg}\;g}^{-1}\right)$	3.22 ± 0.14	3.46 ± 0.40
$R_{adj}^{2}$	0.9554	0.9366
Residual sum of squares	0.03738	0.10
**Pseudo-second-order**		
$k_{S}\;\left({g\;\mathrm{mg}}^{-1}{\;\min}^{-1}\right)$	0.010 ± 0.001	0.003 ± 0.002
$q_{e}\left({\mathrm{mg}\;g}^{-1}\right)$	3.99 ± 0.19	5.01 ± 1.05
$h_{0}\left({\mathrm{mg}\;g}^{-1}\min^{-1}\right)$	0.15 ± 0.01	0.08 ± 0.01
$R_{adj}^{2}$	0.9776	0.9197
Residual sum of squares	0.02	0.12
**Pseudo-general-order**		
$k_{G}\;\left(\min^{-1}\right)$	0.009 ± 0.001	0.006 ± 0.003
$q_{e}\left({\mathrm{mg}\;g}^{-1}\right)$	4.03 ± 0.17	4.56 ± 0.81
$n_{}$	2.05 ± 0.01	1.72 ± 0.01
$R_{adj}^{2}$	0.9870	0.9394
Residual sum of squares	0.02	0.10
**Elovich**		
$\alpha \left({\mathrm{mg}\;g}^{-1}\right)$	0.37 ± 0.04	0.18 ± 0.02
$\beta \left({g\;\mathrm{mg}}^{-1}\right)$	1.20 ± 0.07	0.90 ± 0.07
$R_{adj}^{2}$	0.9845	0.9578
Residual sum of squares	0.02	0.07
**Intraparticle diffusion**		
$k_{id1}\left({\mathrm{mg}\;g}^{-1}\min^{-0.5}\right)$	0.333 ± 0.01	0.556 ± 0.04
*C*_1_ (${\mathrm{mg}\;g}^{-1}$)	0.144 ± 0.07	−1.357 ± 0.20
$R_{adj}^{2}$	0.9928	0.9775
Residual sum of squares	0.04	0.13
$k_{id2}\left({\mathrm{mg}\;g}^{-1}\min^{-0.5}\right)$	0.008 ± 0.001	0.05 ± 0.01
*C*_2_ (${\mathrm{mg}\;g}^{-1}$)	3.17 ± 0.01	2.47 ± 0.06
$R_{adj}^{2}$	----	0.9499
Residual sum of squares	----	0.01

## Data Availability

Not applicable.

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
