# Peer review of "Banana Peel Powder Biosorbent for Removal of Hazardous Organic Pollutants from Wastewater"

_toxics, 2023, doi:10.3390/toxics11080664_

Round 1

Reviewer 1 Report

Major revisions are required and the comments are given below.

1. What is the active component of BPP for adsorption? What is the reason for the difference in the performance of the three growth stages of BP?

2. Several times in the manuscript "Error! The Reference source was not found ".

3. Measurement of specific surface area and analysis of pore volume aperture of BPP.

4. The factors affecting the adsorption performance are too simple to consider, please set the temperature, pH, quality of the adsorbent, and other control groups.

5. The author emphasizes the advantages of low-cost BPP, which should also be compared with the adsorption properties of other biomass materials.

6. Lack of fitting curves in each adsorption model, and the adsorption mechanism is not clear enough.

7. The manuscript hardly cited the literature reports in recent years, and the author should have a better understanding of the latest research status in the field of adsorption. Some typical references are suggested to be cited to enrich the content, e.g. Journal of Bioresources and Bioproducts 2022, 7 (2), 109-115.

8. Improve the writing logic of the manuscript. Some references are missing page numbers.

Minor editing of English language is required.

Reviewer 2 Report

Farias et al., investigated the adsorption of several pollutants (methylene blue, glyphosate and atrazine) using biosorbents derived from banana peels. The adsorption/desorption of pollutants is an interesting and widely investigated topic. The approach of the authors is well performed, the article is well written and the applied methodology merits publication in Toxics. However, after reading the manuscript I have some major and minor comments which ought to be addressed before publication:

-    -    “and microorganisms” in Line 35 should be removed, when summarizing examples of toxic chemicals.

-     -   Line 91: “Musa Paradisiaca L.” should be written in italic and “Paradisiaca” without a capital letter.

-   -     Line 95: what is meant with specially processed ?

-      -  Line 106: “%ww” should be “%w/w”

-      -  Do the authors have any idea of the particle size distribution of the BPP after sieving with a 40 mesh sieve? This could also present information regarding the d50 of the BPP particles.

-     -   Please add RPM in Line 137 regarding the centrifugation (in order to remove the biosorbent).

-   -     Are equations 1,2 and 3 not also applicable for the biosorption of glyphosate and atrazine?

-        Lines 172-173: reference error

-     -   The authors assessed the biosorbent material via SEM and FTIR. However, would it not be more logical to assess its (bio-)chemical composition as the banana peels are used as such? A bio-chemical analysis (for instance Van Soest analysis) would reveal more interesting information (cellulose, hemicellulose, lignin, extractives, etc).

-     -   The BPP are used as such in a non-stable form in contrast to for example biochar. This surely has severe limitations. The chemical structure (hemicellulose, cellulose, lignin) will change over time depending on the application envisaged. This will surely also influence the sorption process. Did the authors took this in consideration?

-     -   Line 222 and 224, there is a line under “°”.

-     -   The authors state the following (Lines 253-255): “minor trend is present with a higher adsorption ability for the semi-ripe BPP, reaching approximately 66% of MB removal after 60 min.” However, did the authors statistically determine whether the adsorption curves are significantly different (Figure) ?

-   -     Please mind significant numbering throughout the text and be consistent (for example Lines 273 and 274, but not limited to this example).

-   -     Can the authors determine the standard error on the obtained kinetic parameters (i.e., k’s and q’s, etc)?

-     -   Line 394, replace “worst”.

-    -    Line 406: “Musa Paradisiaca L.” should be written in italic and “Paradisiaca” without a capital letter.

-   -     It would be interesting to have some more information of the pores (for instance via BET analysis) of the applied BPP.

Reviewer 3 Report

After incorporating the following comments this study may consider for publication:

1.    The study lacks a comprehensive analysis of the potential drawbacks or limitations associated with using banana peel powder (BPP) as a biosorbent. It would be valuable to explore any potential adverse effects or challenges that may arise during the application of BPP in real-world wastewater treatment scenarios.

2.    The experimental setup seems to focus primarily on synthetic solutions spiked with contaminants, which may not fully capture the complexity of real wastewater samples. It would be beneficial to conduct further experiments using actual wastewater samples to evaluate the performance and effectiveness of BPP in a more realistic setting.

3.    The study lacks detailed information on the stability and reusability of the BPP biosorbent. It would be valuable to investigate the potential for regeneration and reuse of BPP, as this information is crucial for evaluating its long-term viability and cost-effectiveness in large-scale wastewater treatment processes.

4.    The mechanisms underlying the sorption processes of methylene blue (MB), atrazine, and glyphosate on BPP should be further elucidated. Additional characterization techniques, such as spectroscopic analysis, could provide more in-depth insights into the specific interactions between the pollutants and the BPP surface.

5.    The study mainly focuses on the biosorption performance of BPP, but it lacks a comprehensive assessment of its overall environmental impact. The authors should consider conducting a life cycle assessment to evaluate the potential environmental burdens associated with the production, utilization, and disposal of BPP as a biosorbent.

6.    The limited range of pollutants tested in the study raises questions about the broader applicability of BPP as a biosorbent. It would be beneficial to investigate its performance with a wider range of contaminants commonly found in wastewater, including heavy metals and other organic pollutants.

7.    The statistical analysis of the experimental data is not adequately addressed in the study. Including statistical methods and presenting the results with appropriate error bars or confidence intervals would enhance the reliability and robustness of the findings.

8.    The study does not provide a comprehensive economic analysis of using BPP as a biosorbent compared to other existing treatment technologies. It would be valuable to include a cost-benefit analysis to determine the economic feasibility and competitiveness of BPP in relation to alternative methods.

9.    The potential challenges associated with the scalability and implementation of BPP in large-scale wastewater treatment facilities are not adequately addressed. It is important to consider the practicality and technical requirements of using BPP on a larger scale, including issues related to process optimization and equipment compatibility.

10. The study does not thoroughly discuss the potential regulatory and safety considerations associated with the use of BPP as a biosorbent. It is important to address any potential concerns related to the release of pollutants or by-products during the biosorption process and ensure that the treated wastewater meets relevant regulatory standards.

Quality of English Language is OK

Round 2

Reviewer 1 Report

The manuscript could be accepted now.

Author Response

We would like to thank the Reviewer for the feedback and comments on our manuscript. We appreciate the suggestions and comments, which raised important issues to be adjusted/improved for the final version of the manuscript. 
Thanks for your favorable final decision.

Reviewer 2 Report

The authors addressed the comments raised by the reviewer and therefore the reviewer accepts this manuscript for publication in Toxics.

Author Response

(The authors gave the same response as above.)

Reviewer 3 Report

The author's response to the queries raised by us seems to be inadequate. Many of the queries are dismissed as being beyond the scope of the study. However, it is important for the author to address and answer these queries in order to improve the readability and quality of the manuscript. By properly addressing the reviewers' queries, the author can provide a more comprehensive and thorough analysis of the study's findings. Taking the time to address these queries will not only enhance the manuscript but also demonstrate the author's commitment to addressing the concerns and feedback of the reviewers. 

Quality of English Language is OK

Author Response

Dear Reviewer,

We would like to express our appreciation for your careful review of our manuscript. We acknowledge the significance of the peer review process in ensuring the quality and integrity of scientific publications and its crucial role in advancing scientific knowledge globally. We understand the importance of preventing the dissemination of flawed works that could impede the progress of young researchers and hinder scientific advancements that directly impact society.

We have made a sincere effort to address all the questions and concerns raised by the reviewer to the best of our abilities, within the limitations of our study and its scope. We apologize if we were unable to address certain broader and more complex issues, such as "life cycle assessment," "performance with a wider range of contaminants, including heavy metals and other organic pollutants," "economic analysis of using BPP as a biosorbent compared to other existing treatment technologies," "economic feasibility and competitiveness," "process optimization and equipment compatibility," and "potential regulatory and safety considerations". We acknowledge that these are important aspects that would provide valuable insights into the practical applications of our findings.

However, we also recognize that science advances step by step, where each contribution brings new nuances and ideas for future work that ultimately culminate in important discoveries that directly impact human life. We firmly believe that the current version of our manuscript contributes significant scientific value to the research community. Our study represents the first application of BPP for the adsorption of glyphosate, the most widely used pesticide worldwide. Moreover, we have systematically investigated the efficacy and kinetics of adsorption, comparing different stages of BPP maturation as well as different pesticides and a dye. These findings provide valuable insights into the potential use of BPP as a biosorbent for pesticide removal.

We thank the reviewer once again for their constructive feedback, which has undoubtedly enhanced the quality of our manuscript. We are confident that our work contributes to the scientific community's understanding of the subject matter and serves as a foundation for future research endeavors.

Sincerely,
Authors

Round 3

Reviewer 3 Report

Editor may accept the manuscript.

Quality of English Language is OK